# Imaging non-collinear antiferromagnetic textures via single spin relaxometry

Aurore Finco [1], Angela Haykal [1], Rana Tanos[1], Florentin Fabre[1], Saddem Chouaieb[1], Waseem Akhtar[1,2], Isabelle Robert-Philip[1], William Legrand[3], Fernando Ajejas [3], Karim Bouzehouane [3], Nicolas Reyren [3], Thibaut Devolder[4], Jean-Paul Adam[4], Joo-Von Kim [4], Vincent Cros [3] & Vincent Jacques [1✉]

Antiferromagnetic materials are promising platforms for next-generation spintronics owing to their fast dynamics and high robustness against parasitic magnetic fields. However, nanoscale imaging of the magnetic order in such materials with zero net magnetization remains a major experimental challenge. Here we show that non-collinear antiferromagnetic spin textures can be imaged by probing the magnetic noise they locally produce via thermal populations of magnons. To this end, we perform nanoscale, all-optical relaxometry with a scanning quantum sensor based on a single nitrogen-vacancy (NV) defect in diamond. Magnetic noise is detected through an increase of the spin relaxation rate of the NV defect, which results in an overall reduction of its photoluminescence signal under continuous laser illumination. As a proof-of-concept, the efficiency of the method is demonstrated by imaging various spin textures in synthetic antiferromagnets, including domain walls, spin spirals and antiferromagnetic skyrmions. This imaging procedure could be extended to a large class of intrinsic antiferromagnets and opens up new opportunities for studying the physics of localized spin wave modes for magnonics.

[1] Laboratoire Charles Coulomb, Université de Montpellier and CNRS, 34095 Montpellier, France. [2] Department of Physics, JMI, Central University, New Delhi, India. [3] Unité Mixte de Physique, CNRS, Thales, Université Paris-Saclay, 91767 Palaiseau, France. [4] Centre de Nanosciences et de Nanotechnologies, CNRS, Université Paris-Saclay, 91120 Palaiseau, France. ✉email: vincent.jacques@umontpellier.fr

**D**espite decades of continuous progress in magnetic microscopy and the assets of antiferromagnetic materials for spintronics[1,2], nanoscale imaging of the magnetic order in antiferromagnets still remains a notoriously difficult task[3]. A first strategy consists in using probe particles, like photons or electrons, whose properties are modified by their interaction with the sample magnetization. This approach can be pursued with several techniques including X-ray photoemission electron microscopy[4,5] and spin-polarized scanning tunneling microscopy[6,7], the latter providing direct imaging of the Néel order with atomic scale resolution. Yet, these highly complex techniques require either synchrotron-based experimental facilities or conductive samples with nearly perfect surfaces in ultrahigh vacuum. A more versatile approach to magnetic microscopy relies on the detection of static magnetic fields generated outside the sample. In antiferromagnets, weak magnetic fields are produced by the rotation of tiny uncompensated moments whose orientations are linked to the Néel order. Mapping such fields requires the use of scanning magnetometers combining high field sensitivity with nanoscale spatial resolution. Besides magnetic force microscopy (MFM) pushed at its limits[8,9], these performances are offered by a new generation of table-top magnetic microscopes employing a single Nitrogen-Vacancy (NV) defect in diamond as a non-invasive, atomic-size quantum sensor[10–12]. This technique was recently used to image the stray field distribution produced by spin textures in magnetoelectric antiferromagnets under ambient conditions[13–15]. However, it remains limited to the study of a special class of antiferromagnets producing strong enough static magnetic fields.

In this paper, following a recent proposal[16], we demonstrate that antiferromagnetic spin textures can be imaged by sensing the magnetic noise they locally produce, rather than their static magnetic stray fields. To this end, we employ a single NV defect in diamond for nanoscale noise sensing. Magnetic noise with a spectral component resonant with the electron spin transition of the NV defect is usually detected by recording variations of its longitudinal spin relaxation time $T_1$[17]. This method, commonly referred to as relaxometry[18], has been used for various purposes in the past years, including the study of Johnson noise in metals[19,20], current fluctuations in graphene devices[21], paramagnetic nanoparticles[22–24] and spin waves in ferromagnets[25–27]. In contrast to these previous works, we introduce a relaxometry-based imaging mode, which relies on the simple measurements of the photoluminescence (PL) signal of the NV defect under continuous laser illumination. The efficiency of this method is demonstrated by imaging spin textures in synthetic antiferromagnets (SAFs), including domain walls, spin spirals and antiferromagnetic skyrmions.

## Results

**Principle of the measurement**. The imaging mechanism is illustrated by Fig. 1. The spin-dependent PL response of the NV defect is modeled by considering a closed two-level system corresponding to the ground states with spin projection $m_s = 0$ (bright state) and $m_s = \pm 1$ (dark state)[28] (Fig. 1a). Under continuous laser illumination, the PL intensity is linked to steady-state spin populations, which only result from the competition between the spin relaxation rate $\Gamma_1 = \frac{1}{2T_1}$ and optically-induced spin polarization in the $m_s = 0$ sublevel with a rate $\Gamma_p$ that depends on the optical excitation power. Within this simple model, the evolution of the PL intensity with the spin relaxation time $T_1$ is shown in Fig. 1b for two different optical powers (see Supplementary Note 2). Under standard experimental conditions corresponding to $\Gamma_1 \ll \Gamma_p$, the NV defect is efficiently polarized in $m_s = 0$ by optical pumping, leading to a high PL signal.

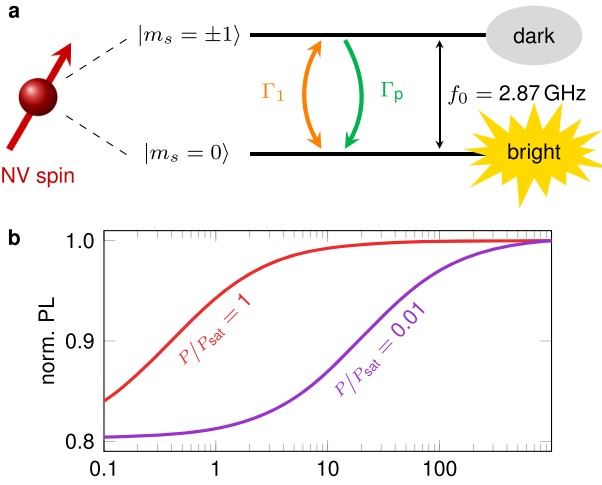

**Fig. 1 Principle of the measurement. a** Simplified two-level system used to model the spin-dependent photoluminescence (PL) of the NV defect. **b** Normalized PL intensity of the NV defect as a function of the spin relaxation time $T_1$ for two different regimes of optical excitation power $P$. The parameter $P_{\text{sat}}$ denotes the saturation power of the NV defect optical transition.

However, if magnetic noise increases the spin relaxation rate such that $\Gamma_1 \sim \Gamma_p$, NV spin polarization is degraded and the overall PL intensity is reduced. Magnetic noise is therefore directly imprinted into the PL signal of the NV defect. In addition, depending on the optical excitation power, the PL drop occurs at different spin relaxation times (Fig. 1b), so that the laser power can be adjusted to optimize the imaging contrast. In the following, we show that this procedure can be employed for nanoscale, all-optical imaging of non-collinear antiferromagnetic spin textures by probing the magnetic noise they locally produce via thermally-activated spin waves[16].

**Experimental details**. To demonstrate the efficiency of the method, we image the magnetic order in synthetic antiferromagnets (SAFs). These magnetic systems are made of thin ferromagnetic layers coupled antiferromagnetically through Ruderman–Kittel–Kasuya–Yoshida (RKKY) interlayer exchange coupling. The high tunability of magnetic parameters in SAFs offers many opportunities for antiferromagnetic spintronics[29], from efficient spin-torque induced domain wall motion in racetrack memory devices[30] to the recent stabilization of antiferromagnetic skyrmions[8]. The SAF structure used in this work consists of a sputtered [Pt/Co/Ru]$_{\times 2}$ multilayer stack with broken inversion symmetry, in which the non-magnetic Ru-Pt spacer layer ensures RKKY-based antiferromagnetic coupling between the two identical Co layers (see Fig. 2a). In such samples, the Pt/Co interface provides a sizable interfacial Dzyaloshinskii–Moriya interaction (DMI) combined with an effective perpendicular magnetic anisotropy, that can be finely tuned by varying the Co thickness $t_{\text{Co}}$[8] (see Supplementary Fig. 1). Magnetic imaging is performed with a scanning-NV magnetometer operating under ambient conditions. A commercial diamond tip hosting a single NV defect at its apex (Qnami, Quantilever MX) is integrated into an atomic force microscope and scanned above the SAF surface (see Fig. 2a). At each point of the scan, a confocal optical microscope is used to record the PL signal of the NV defect under continuous optical illumination with a green laser. For the experiments described below, the flying distance of the NV spin sensor is $d_{\text{NV}} = 79 \pm 5$ nm, as measured

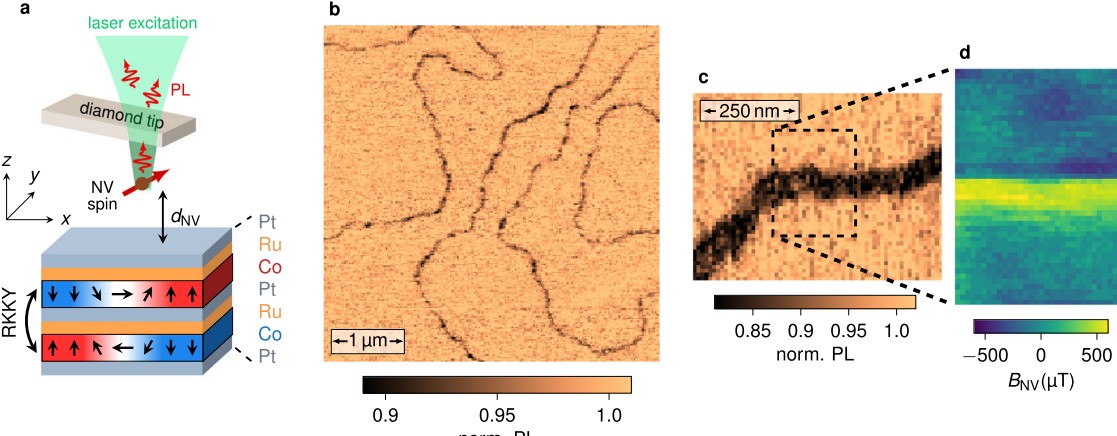

**Fig. 2 All-optical imaging of domain walls in a synthetic antiferromagnet. a** Simplified sketch of the scanning-NV magnetometer and the SAF structure. The two magnetic Co layers are antiferromagnetically coupled through the Ru/Pt spacer layer by RKKY exchange interaction. The exact composition of the stack is Ta(10)/Pt(8)/Co($t_{Co}$)/Ru(0.75)/Pt(0.6)/Co($t_{Co}$)/Ru(0.75)/Pt(3), with the thicknesses in nm. The effective perpendicular magnetic anisotropy of the SAF is controlled by varying the Co thickness $t_{Co}$. **b** PL quenching image recorded with the scanning-NV magnetometer above a SAF sample with perpendicular magnetic anisotropy ($t_{Co} = 1.41$ nm), showing an overview of the magnetic domain structure. **c** Closer view of a domain wall. **d** Static magnetic field distribution recorded above the area marked with the dashed rectangle in **c**. Only the component of the stray field along the NV quantization axis, $B_{NV}$, is measured. This axis is defined by the spherical angles ($\theta_{NV} = 58°$, $\phi_{NV} = 103°$) in the laboratory frame of reference ($x$, $y$, $z$).

through an independent calibration procedure[31] shown in Supplementary Fig. 2.

**Imaging domain walls in a SAF via single spin relaxometry.** We first focus on a SAF structure with a significant effective perpendicular magnetic anisotropy, and thus large antiferromagnetic domains. A typical PL map recorded with scanning-NV magnetometry is shown in Fig. 2b. It reveals a network of sharp PL quenching lines, which correspond to domain walls in the SAF. We note that very similar PL maps can be obtained above ultrathin ferromagnetic samples producing sufficiently strong magnetic fields (>5 mT) to induce an efficient mixing of the NV defect's spin sublevels[32–34]. In a SAF, however, the static field generated above a domain wall is too weak to produce such a spin mixing, since this field mainly results from the slight vertical shift between the two compensated magnetic layers. This is confirmed by quantitative magnetic field imaging above a domain wall (Fig. 2c, d). For this measurement, a microwave excitation is combined with optical illumination to record the Zeeman-shift of the electron spin resonance (ESR) frequency of the NV defect at each point of the scan[12]. The maximum stray field above the domain wall is around 500 µT, in agreement with the value expected from micromagnetic simulations (see Supplementary Fig. 7). Such a static field is not strong enough to induce a detectable PL quenching[32]. Furthermore, this control experiment indicates that variations of the PL signal are localized at domain walls in the SAF sample, and therefore cannot be explained by surface-induced energy transfer effects[35] or charge-state conversion of the NV spin sensor[36].

In our experiments, PL quenching is rather induced by magnetic noise localized at SAF domain walls. This is verified through measurements of the longitudinal spin relaxation time $T_1$ of the NV defect using the experimental sequence shown in Fig. 3a. The NV defect is first initialized into the $m_s = 0$ spin sublevel with a laser pulse. After relaxation in the dark during a time $\tau$, a second laser pulse is used to read-out the final population in $m_s = 0$ by recording the spin-dependent PL signal[37]. As illustrated in Fig. 3b–e, the measurement is performed for three different positions of the NV spin sensor. When the tip is retracted far away from the sample surface, we obtain $T_1 = 860 \pm 300$ µs, a value smaller than the one commonly

inferred for single NV defects in bulk diamond samples. We attribute this observation to magnetic noise generated by paramagnetic impurities lying on the diamond tip surface[37]. By engaging the tip above an antiferromagnetic domain, the spin relaxation time drops to $T_1 = 120 \pm 10$ µs, indicating the presence of an additional source of magnetic noise with a spectral component at the ESR frequency of the NV defect. We checked that this noise comes from the magnetization of the sample by performing control experiments with the tip engaged on non-magnetic surfaces (see Supplementary Fig. 8). Finally, when the tip is placed above a SAF domain wall, $T_1$ is further shortened down to $22 \pm 2$ µs, revealing that magnetic noise is stronger at domain walls. As shown in Fig. 1b, a large PL quenching contrast is indeed expected at low optical excitation power when the $T_1$ time drops from 120 µs to 20 µs. In addition, the quenching contrast should almost vanish when the excitation power reaches the saturation of the NV defect optical transition. Magnetic images recorded at different optical powers confirm this effect (Fig. 3f–h), illustrating that the imaging contrast can be optimized by adjusting the excitation power.

**Origin of the magnetic noise.** The observed variations in NV spin relaxation time are explained by magnetic noise generated by thermal populations of magnons. This is supported by micromagnetic simulations of the dispersion relation of spin waves and their power spectral density (see Supplementary Note 3). In Fig. 4a we show the simulation geometry, which consists of an equilibrium magnetic configuration comprising Néel-type domain walls with an antiparallel alignment between the layers. In the uniform SAF domain and in the absence of applied magnetic fields, the dipolar coupling between the two ferromagnetic layers lifts the degeneracy between the acoustic and optic spin wave modes, as indicated by the solid lines in Fig. 4b. Propagation of spin waves is characterized by a quadratic dispersion in the wave vector $k$ with a frequency gap determined by the perpendicular magnetic anisotropy and the RKKY interlayer exchange coupling. By including the measured value of the Gilbert damping ($\alpha \sim 0.1$), however, these two branches are smeared out as shown in the color map of the simulated power spectral density in Fig. 4b. The bottom of the spin wave band, which is determined by the frequency of the uniform mode ($k = 0$),

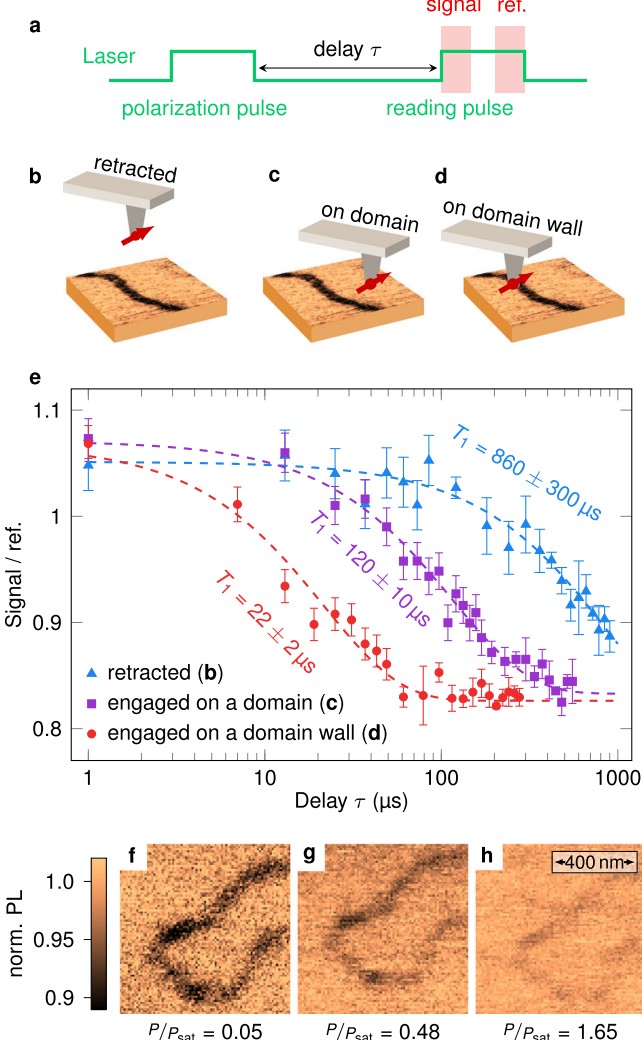

**Fig. 3 Measurement of the NV spin relaxation time. a** Experimental sequence used to measure $T_1$. The spin-dependent PL signal is integrated at the beginning of the reading pulse (300 ns window), and normalized using a reference PL value obtained at the end of the pulse[37]. **b–d** Sketches of the position of the NV sensor with respect to the magnetic structure. The tip is either **b** retracted, **c** engaged above a domain, or **d** engaged above a domain wall. **e** NV spin relaxation curves measured for the three different tip positions. The $T_1$ time is obtained from a fit to an exponential decay (dashed lines). **f–h** PL quenching images recorded above the same domain wall with increasing excitation laser power $P$. In these experiments the saturation power of the NV defect optical transition is $P_{sat} = 450\ \mu W$. For the images shown in panels **f–h**, the values of the PL rate above the uniform domain (bright area) are $7 \times 10^3\ s^{-1}$, $50 \times 10^3\ s^{-1}$ and $100 \times 10^3\ s^{-1}$, respectively. The acquisition time per pixel is 150 ms for all measurements.

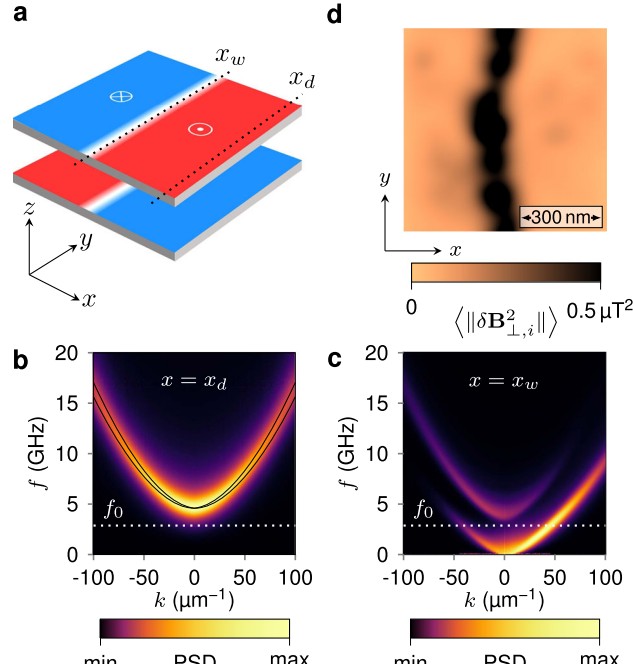

**Fig. 4 Origin of the magnetic noise. a** Equilibrium domain wall configuration obtained by micromagnetic simulation. $x_w$ and $x_d$ indicate lines along which spin wave propagation is characterized. **b** Dispersion relation of the propagating spin wave modes in the SAF domain, taken at $x = x_d$ in **a**. The solid lines indicate a theoretical prediction, while the color map represents the power spectral density (PSD) computed from the damped transient response in micromagnetic simulations. **c** Dispersion relation of the channeled domain wall spin waves computed at $x = x_w$ in **a**. The white dashed lines in **b**, **c** indicate the position of ESR frequency of the NV defect at zero field, $f_0 = 2.87$ GHz. **d** Simulated map of the magnetic noise intensity at the frequency $f_0$, computed at a flying distance $d_{NV} = 80$ nm above a domain wall configuration in the presence of disorder. The quantity $\langle \|\delta\mathbf{B}_{\perp,i}\|^2 \rangle$ denotes the noise intensity perpendicular to the NV quantization axis, which is obtained from an average over 500 different realizations of the random driving field.

remains above the ESR frequency of the NV defect at zero field, $f_0 = 2.87$ GHz [Fig. 4b]. However, given the line broadening induced by damping, the NV frequency falls into the tail of the thermal power spectrum (see Supplementary Fig. 6), which results in a reduction in $T_1$ time when the tip is engaged above a uniform SAF domain. For a domain wall, on the other hand, we find as in the ferromagnetic case[38,39] a gapless mode that propagates parallel to the domain wall, with a nonreciprocity that arises from a combination of the Dzyaloshinskii–Moriya and dipolar interactions[40,41] (Fig. 4c). These modes have been recently observed experimentally with time-resolved scanning transmission X-ray microscopy (STXM)[42]. To illustrate how the

existence of such gapless modes provides the spatial contrast in NV-based relaxometry, we compute a spatial map of the magnetic noise intensity at $f_0 = 2.87$ GHz in the presence of a domain wall in a realistic, disordered SAF system. To achieve this, we evoke the fluctuation-dissipation theorem; rather than computing the noise power spectrum directly from the magnetic response to fluctuating thermal fields, which represents a Langevin dynamics problem that is computationally intensive, we calculate instead the harmonic response of the magnetization dynamics to spatially random fields at the frequency $f_0$ (see Supplementary Note 3). Figure 4d represents the resulting ensemble-averaged map of the magnetic noise intensity at the flying distance $d_{NV}$ of the NV defect. As observed in the experimental data, the strongest noise intensity is obtained at the domain wall, due to the thermally-excited gapless modes. We note that the NV defect is mainly sensitive to magnetic excitations with a wave vector on the order of the inverse of the flying distance. More precisely, this filter in $k$-space has a shape given by[25] $k^2 e^{-2kd_{NV}}$. For the experiments shown in Fig. 3, $d_{NV} \sim 80$ nm and the NV center is therefore mostly sensitive to thermal magnons with wave vectors lying between 5 and 30 $\mu m^{-1}$. From the dispersion relation in Fig. 4c, it appears that we are mainly detecting excitations which are slightly off the maximum power density, but nevertheless sufficient to obtain a significant imaging contrast.

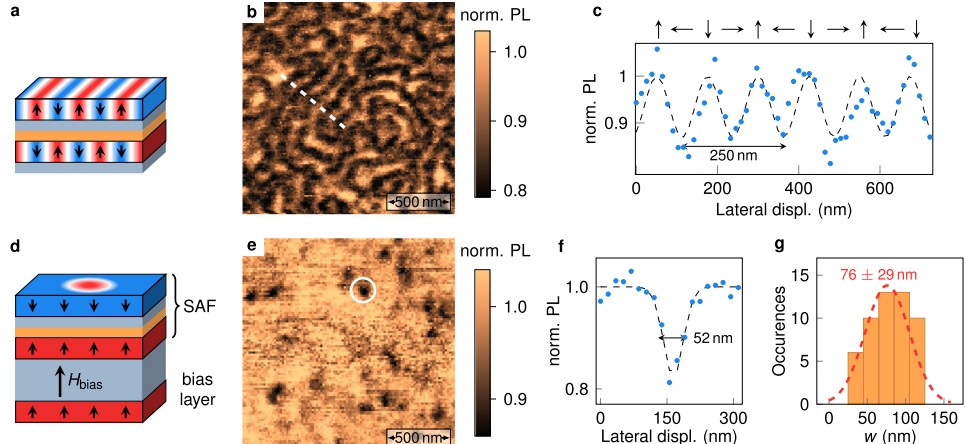

**Fig. 5 Imaging antiferromagnetic spin spirals and skyrmions via single spin relaxometry. a** Sketch of a SAF sample hosting a spin spiral. The composition of the stack is identical to the one shown in Fig. 2a, with a Co thickness ($t_{Co} = 1.47$ nm) leading to a SAF with vanishing magnetic anisotropy. **b** PL quenching image of a spin spiral state. **c** Line profile along the white dashed line in **b**. **d** Sketch of a SAF sample including an additional bias layer which enables the transformation of the spin spiral into antiferromagnetic skyrmions. The composition of the bias layer is Ta(10)/Pt(8)/[Co(0.6)/Pt(0.45)]₃. **e** PL quenching image showing antiferromagnetic skyrmions in the biased-SAF sample. **f** Line profile across the skyrmion marked with the circle in **e**. The dashed line is data fitting with a Gaussian function. **g** Statistical distribution of the full width at half maximum $w$ of the PL quenching spots, inferred from measurements over a set of 52 isolated skyrmions. The red dashed line is a fit with a normal distribution.

**Imaging spin spirals and skyrmions in a SAF**. So far, we have shown that all-optical NV-based relaxometry provides nanoscale imaging of domain walls in a SAF with perpendicular magnetic anisotropy. We now go a step further by demonstrating that this simple method can also be used to image more complex textures, such as antiferromagnetic spin spirals and skyrmions[8]. For this purpose, we exploit the high tunability of magnetic parameters in SAF systems. Keeping the very same stack structure, the Co thickness is first carefully adjusted in order to obtain a SAF with vanishing effective magnetic anisotropy (Fig. 5a). In this case, the magnetic ground state corresponds to an antiferromagnetic spin spiral, whose period is given by $\lambda = 4\pi\frac{A}{D}$, with $A$ the Heisenberg exchange parameter and $D$ the effective DMI constant. A typical PL quenching image recorded with the scanning-NV sensor above a SAF with vanishing magnetic anisotropy is shown in Fig. 5b. A disordered spin spiral structure is clearly visible through alternating bright and dark PL stripes. As for the domain wall case, micromagnetic simulations confirm that the magnetic noise intensity at $f_0$ is stronger in the part of the spin spiral in which the magnetization lies in the film plane, which provides the imaging contrast in NV-based relaxometry (see Supplementary Fig. 4). Analysis of line profiles of the PL quenching map indicates a spin spiral period $\lambda \sim 250$ nm (Fig. 5c). Considering an exchange parameter $A = 20$ pJ m$^{-1}$, this period corresponds to an effective DMI constant $D = 1$ mJ m$^{-2}$, in good agreement with earlier studies[8].

Starting from a spin spiral state, antiferromagnetic skyrmions can then be stabilized by using a bias interaction[8]. This is realized by adding a uniformly magnetized bias layer coupled ferromagnetically to the bottom layer of the SAF with vanishing anisotropy. This coupling creates an effective out-of-plane bias field $\mu_0 H_{bias}$ (Fig. 5d), which plays the same role as an external magnetic field for the stabilization of magnetic skyrmions in ferromagnetic materials. A PL quenching image recorded above a biased-SAF is shown in Fig. 5e. It features isolated PL quenching spots which correspond to antiferromagnetic skyrmions. A statistical analysis of line profiles over a set of 52 isolated skyrmions indicate an average width at half maximum of about 76 nm (Fig. 5f), which is likely limited by the flying distance of the NV spin sensor. These results are in line with MFM measurements performed under vacuum on similar samples[8].

Despite the fact that the internal breathing mode frequencies of the skyrmion are larger than $f_0$, the calculated noise map at this frequency indicates a stronger intensity above skyrmions (see Supplementary Fig. 5). We attribute this to the scattering of propagating spin waves by the skyrmions, which results in small displacements and deformations of their equilibrium profiles.

## Conclusion

The present work introduces a new way to image non-collinear antiferromagnetic spin textures with nanoscale spatial resolution, which relies on the detection of magnetic noise locally produced by thermal populations of magnons. This is achieved by adding a relaxometry-based imaging mode to the scanning-NV magnetometry toolbox, which is based on measurements of variations in the NV defect PL signal induced by magnetic noise. Beyond ordered antiferromagnetic structures like domain walls, spirals, and skyrmions, this imaging procedure could be extended to study magnetic order and disorder in other low-moment materials, such as domain structures in two-dimensional van der Waals systems with low Curie temperature in which spin fluctuations would become dominant under ambient conditions. The sensitivity to magnetic noise with nanoscale spatial resolution also opens up possibility of mapping out magnon-driven spin currents in different magnetic textures, which would provide valuable insights into magnon transport that are inaccessible through other experimental means.

## Data availability

The data[43] that support the findings of this study are available in Zenodo (https://zenodo.org/record/4310011) with the identifier 10.5281/zenodo.4310011.

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

## Acknowledgements

The authors thank J.-P. Tetienne for fruitful discussions. This research has received funding from the DARPA TEE program, the European Union H2020 Program under Grant Agreement No. 820394 (ASTERIQs) and No. 824123 (SKYTOP), the European Research Council under Grant Agreement No. 866267 (EXAFONIS), the French Agence Nationale de la Recherche through the project TOPSKY (Grant No. ANR-17-CE24-0025). A.F. acknowledges financial support from the EU Horizon 2020 Research and Innovation program under the Marie Sklodowska-Curie Grant Agreement No. 846597 (DIMAF).

## Author contributions

V.C. and V.J. conceived and coordinated the project. A.F., A.H., R.T., F.F., S.C., W.A., I.R. and V.J. conducted and analyzed the NV experiments. W.L., F.A., K.B., N.R., V.C. and T.D. designed, prepared and characterized the samples. J.-P.A. and J.-V.K. performed the analytical modeling and micromagnetics simulations. A.F., I.R., J.-V.K. and V.J. wrote the manuscript. All the authors discussed the data and commented the manuscript.

## Competing interests

The authors declare no competing interests.
