## [Peer Review File · Nature Communications]

Reviewers' Comments:

Reviewer #1:

Remarks to the Author:

This manuscript presents a study on the use of commercial diamond (NV) nanoscale magnetometer tips for the measurement of magnetic noise in antiferromagnetic thin-film samples. The novelty in this study is the use of relaxometry ('T1') sensing to detect magnetic fluctuations, rather than static fields, in antiferromagnetic materials. This is an experimental realisation of a theoretical proposal and in this case the demonstration reveals domain walls, spin spirals and skyrmions in one particular synthetic antiferromagnetic (SAF) system. The paper is extremely well written, with a very supportive set of supplementary information, showing good understanding and characterisation of this new technique. It is likely to be of very high interest in the SAF field and after consideration of my technical comments I would recommend it for publication.

In terms of suitability of this manuscript for Nature Communications, it is my opinion that it is borderline. However, on balance I would suggest that it is suitable if the authors satisfactorily address the concerns listed below. This takes into consideration the following positive and negative 'impact' aspects of this work:

- a. (Positive) Nanoscale imaging of magnetic order in SAF systems is an important/interesting technological and scientific goal. This paper describes one way to achieve this imaging, with a well established understanding of the underlying imaging mechanism.
- b. (Negative) The use of T1/relaxometry in NV-diamond nanoscale imaging is not at all new, and the implementation of this scheme in this work is rudimentary, not taking into account any of the potential pitfalls of the continuous-laser implementation of this technique.
- c. (Negative) The use of NV-diamond magnetometry more broadly is also not new in thin-film magnetic systems, and this work fails to observe a new feature of such systems. Specifically, this work is nominally aimed at the measurement of domain walls (and other features) which exhibit net-zero magnetization. I can find neither evidence of such a measurement here, nor an analysis of the presented data which could be used to predict such measurements occurring in the future.

In terms of specific concerns and/or potential areas for improvement:

1. On page 4 and in Figure 1, the authors describe the need to optimize the laser power, as this affects the polarisation rate of the NV centre. As such, the continuous-laser approach in this work is better suited to magnetic features which exhibit higher magnetic noise power densities, allowing higher laser powers and higher subsequent photoluminescence (PL) intensities. This would suggest that the NV system in this implementation is actually progressively less sensitive when used to image weaker (oscillating) magnetic features. There doesn't appear to be any analysis of what sensitivity is required here, and whether this technique is likely to be superior to a more traditional pulsed-laser relaxometry measurement technique.
2. The use of PL intensity as a proxy for T1 is only suitable for NV systems where there is no significant extrinsic T1 modification. For near-surface NV centres this is known to be problematic, as charge-trapping/transfer and energy resonance effects (eg. FRET) can also modulate the NV PL signature when an NV is brought into close proximity with a sample surface. Similarly, the reflectivity of the sample can also cause significant PL modulation, if the actual laser intensity at the NV is affected. I suggest that this be recognised in the paper, even if the authors believe it to be a mitigated issue in the NV-tip implantation under review.
3. While the NV-tip system under review is commercially available, it is not clear what is known about the NV-tip distance. More information (at least what is known) should be put in the manuscript, regarding NV orientation, depth from tip and tip radius. This is needed to better understand what is meant by the 'flying distance' of the NV spin sensor (page 5).
4. In Figure 3 (f-g) a series of domain wall images are recorded using different laser powers. Here it is shown that the normalised PL contrast is increased for lower laser powers. However it is not obvious from these images or otherwise whether the actual signal to noise ratio is improved here. With a lower contrast comes a lower PL intensity (therefore signal intensity). Is there a better way

to plot this data, as a proper signal/noise (SNR) analysis? The use of 'normalised PL' in most plots appears to obfuscate this issue. Should not there also be more analysis of SNR in this context throughout the manuscript?

5. In figure 4 the origin of domain wall noise spectra is shown, including the fact that the domain walls exhibit a much lower spread of magnetic noise intensities – towards DC. This suggests to me that an AC technique, such as a decoherence (T_2) imaging mode may give similar if not superior data for these systems. Is this a universal characteristic of these SAF systems and do the authors still believe that relaxometry gives the best imaging modality?

6. One of the apparent limitations of the NV-tip magnetometry in the context of SAF systems is the common need for magnetic field biasing. In this manuscript a zero-bias relaxometry measurement is performed, I assume primarily to avoid perturbing the sample magnetization. However while this is a feature of the measurement, it would also appear to be a significant limitation. Meaning that only ~ 3 GHz magnetic signals can be detected. As this manuscript is intended to pave the way to future such measurements, this limitation should be recognised and discussed. Is it perhaps useful to suggest the development of alternative control of the NV spin transition frequency, such as controlled-strain NV-tip systems?

7. My reading of this manuscript is that there is a strong need for imaging of net-zero magnetization features. However there is no demonstration of such features here, nor a description of the likely sensitivity required to see such features, or what kinds of SAF devices would exhibit such features?

Reviewer #2:

Remarks to the Author:

In this paper, Finco et. al. present single spin relaxometry as a tool to image non-collinear antiferromagnetic textures. Magnetic noise emanating from non-collinear antiferromagnetic textures is detected as a decrease in photoluminescence signal under continuous laser exposure. The efficiency of this method is highlighted by imaging antiferromagnetic textures like domain walls, spin spirals, and antiferromagnetic skyrmions as realized in synthetic antiferromagnets.

The experimental observations presented here corroborates relaxometry-based imaging as a proof-of-concept following a recent theoretical proposal [Physical Review B 98, 180409 (2018)]. The paper will certainly be of interest to researchers working in this field as it extends the use of relaxometry-based imaging to antiferromagnetic materials which are promising for next-generation spintronics.

In regard to the results presented, I believe a few aspects need to be addressed:

*) For the case of domain walls, when the NV is above the uniform domain at x_d in Fig. 4a, the authors suggest that the Gilbert damping value of 0.1 broadens the magnon spectral density in frequency and thus couples to the NV. First, I do not see any justification/measurement that calibrates the value to α to be 0.1. Second, does two-magnon noise play a role in this case - when the NV ESR falls within the magnon spectrum gap as shown in Fig. 4b?

*) For the case of imaging antiferromagnetic skyrmions, the authors suggest that the reason that NV senses the skyrmion (in spite of the fact that its eigenspectrum does not have a mode at 2.87 GHz) by means of scattered magnons against the skyrmion. The supplementary material argues this by evaluating the spinwave dispersion of the synthetic antiferromagnet in the uniform state which does have modes at the NV ESR frequency. The conclusion based on this argument is not very convincing.

The results of this paper are indeed novel and will surely be of interest to the researchers working in this field. However, with proper clarifications provided for the points mentioned above, I do think that this manuscript satisfies the criterion for publication in Nature Communications.

Reviewer #3:

Remarks to the Author:

Finco et al. use optical measurements of the spin-relaxation rate of a scanning NV center in a diamond tip to image domain boundaries in antiferromagnetic structures with nanoscale resolution. Due to a small or absent net magnetization, antiferromagnets are notoriously challenging to image via magnetometry. Finco et al. show that antiferromagnetic domain boundaries in fact generate strong local magnetic noise that can be detected via the NV relaxation rate. They characterize this rate by its impact on the NV photoluminescence, thus avoiding the necessity of using microwaves. This makes it a straightforward method that starts to approach the ease-of-use of the magnetic force microscope (MFM) while going beyond MFM in its ability to detect magnetic fields that fluctuate rapidly (on the GHz scale). They image a variety of spin structures and provide a theoretical explanation for the microscopic origin of the detected magnetic signals. This work pushes the state-of-the-art in NV magnetometry and provides new insight into magnetism. Because it highlights the potential of using NV-magnetometry for general materials characterization, it should be of broad interest in the condensed-matter physics and materials science communities. The data are well presented and the conclusions are supported by the data and their analysis. It should be accepted in Nature communications.

I have some minor comments:

The authors could elaborate on the challenges for extracting T1 quantitatively from photoluminescence maps. In particular since the technique relies on striking a rather precise balance between illumination power (a control parameter) and relaxation time T1 (not a control parameter). This may limit the material systems to which this technique can be applied and extend the time required for calibration measurements. Can the authors comment on the shortest T1 they can determine via this technique? Can the authors comment on the limit in optical power that can be applied? (at some point a high optical excitation power should cause the NV to reside most of the time in the electronic excited state)

It is unclear how the data (in e.g. Fig. 2b) are normalized. Please explain. I also wonder if the normalization procedure is prone to artefact: In particular, it seems to me that a spatial change in background (non-NV) photoluminescence would also give a change in contrast. Can the authors comment on this? And why do the authors plot the ratio and not the difference between the two PL rates?

Since the authors chose a material system in which the stray fields of boundaries are large enough to be imaged by static magnetic field imaging, it is not immediately obvious that PL-based relaxometry can access information that traditional vector magnetometry through ODMR cannot. Can the authors be explicit about what new information can be extracted?

Can the authors comment on the presence of domain structures in Figure 3h? According to the presented theory in Figure 1, shouldn't there be no noticeable difference in PL between $T_1=22 \mu\text{s}$ and $T_1=860 \mu\text{s}$?

A different order of the labels in Figure 4 would be clearer

Authors' Rebuttal to Reviewer Reports – NCOMMS-20-30850-T/Finco

We would like to thank Dr. Bladwell, the editorial staff of *Nature Communications*, and the three Reviewers for considering our manuscript for publication. We are encouraged by the overall positive comments of the three Reviewers. Reviewer #1 suggests the manuscript in its present form is borderline for publication, but this can be rectified by addressing the criticisms raised. Reviewer #2 suggests that publication can be considered after some issues are addressed, while Reviewer #3 recommends publication.

In the following, we include a point-by-point response to all the questions and criticisms raised by the three Reviewers. Their remarks are included verbatim and typeset in *blue italics* for the sake of readability. Changes to the manuscript are highlighted in red in the resubmitted files.

We are confident we have addressed all pertinent remarks and clarified all issues in full. We therefore resubmit our revised manuscript for further consideration in the journal.

Reviewer #1 (Remarks to the Author):

This manuscript presents a study on the use of commercial diamond (NV) nanoscale magnetometer tips for the measurement of magnetic noise in antiferromagnetic thin-film samples. The novelty in this study is the use of relaxometry ('T1') sensing to detect magnetic fluctuations, rather than static fields, in antiferromagnetic materials. This is an experimental realisation of a theoretical proposal and in this case the demonstration reveals domain walls, spin spirals and skyrmions in one particular synthetic antiferromagnetic (SAF) system. The paper is extremely well written, with a very supportive set of supplementary information, showing good understanding and characterisation of this new technique. It is likely to be of very high interest in the SAF field and after consideration of my technical comments I would recommend it for publication.

In terms of suitability of this manuscript for Nature Communications, it is my opinion that it is borderline. However, on balance I would suggest that it is suitable if the authors satisfactorily address the concerns listed below. This takes into consideration the following positive and negative 'impact' aspects of this work:

a. (Positive) Nanoscale imaging of magnetic order in SAF systems is an important/interesting technological and scientific goal. This paper describes one way to achieve this imaging, with a well-established understanding of the underlying imaging mechanism.

We thank the reviewer for this positive assessment of our work. However, we want to underline that the magnetic imaging technique which we demonstrate is not restricted to SAF samples and could be applied to a broad range of magnetic samples hosting non-collinear spin textures.

b. (Negative) The use of T1/relaxometry in NV-diamond nanoscale imaging is not at all new, and the implementation of this scheme in this work is rudimentary, not taking into account any of the potential pitfalls of the continuous-laser implementation of this technique.

We agree with the reviewer that the use of relaxometry for nanoscale imaging is not new, even though the literature on this topic remains scarce. To the best of our knowledge, the published work about scanning relaxometry at the nanoscale is so far limited to 4 examples:

- i) Pellicione *et al*, Phys. Rev. Appl. 2, 054014 (2014);
- ii) Schmid-Lorch *et al*, Nano Lett. 15, 4942 (2015);
- iii) Tetienne *et al*, Nano Lett. 16, 326 (2016);
- iv) Ariyaratne *et al*, Nat. Commun. 9, 2406 (2018).

Furthermore, we note that scanning-relaxometry with a single, scanning NV defect was only demonstrated in iv).

The papers i), ii) and iii) report on the detection of paramagnetic particles. In these works, the acceleration of the NV spin relaxation rate is probed by recording the PL intensity while applying laser pulses separated by a fixed time delay τ , whose

value is carefully adjusted to obtain a good signal to noise ratio. Such a single- τ imaging procedure is qualitative, since it does not allow the value of T_1 to be inferred at each point of the scan, like in our work. In article iv), quantitative maps of the T_1 relaxation time are measured from which the local variations of electrical conductivity in nanopatterned metallic conductors are inferred, through the detection of Johnson noise. Here, the authors introduce a clever measurement procedure combining adaptive single- τ pulse sequences with spin-to-charge readout of the NV spin sensor, which enables to increase significantly the imaging speed.

The main novelty of our work is the clear demonstration that NV-based relaxometry can be used for all-optical imaging of complex non-collinear spin textures in antiferromagnetic systems, which are currently attracting widespread scientific interest for the design of novel spintronic devices. This is realized by recording the variations of T_1 through a new mechanism, the variation of the emitted PL under continuous laser illumination, which is practically and conceptually very simple to implement. We do not see any potential pitfalls of this method compared to other scanning-relaxometry measurement protocols.

c. (Negative) The use of NV-diamond magnetometry more broadly is also not new in thin-film magnetic systems, and this work fails to observe a new feature of such systems. Specifically, this work is nominally aimed at the measurement of domain walls (and other features) which exhibit net-zero magnetization. I can find neither evidence of such a measurement here, nor an analysis of the presented data which could be used to predict such measurements occurring in the future.

NV magnetometry is indeed becoming an important tool to probe the physics of magnetic thin films. However, our work is the first experimental implementation of the idea of identifying non-collinear spin textures through the detection of the magnetic noise they locally generate. We believe that this approach could be particularly useful for studying magnetic order in compensated systems, like pure antiferromagnets, where stray fields are vanishingly small but thermal spin waves are amply present under ambient conditions. In our work, we want to provide a robust proof-of-concept illustrating the potential of this novel imaging mode. As a result, we have chosen a magnetic system - the synthetic antiferromagnet - on which we are also able to achieve standard magnetometry measurements (static fields) as a control experiment (see Fig. 2). We note that the existence of gapless spin wave modes localized at non-collinear spin textures is a common feature of numerous magnetic materials, making the imaging procedure very general.

In terms of specific concerns and/or potential areas for improvement:

1. On page 4 and in Figure 1, the authors describe the need to optimize the laser power, as this affects the polarisation rate of the NV centre. As such, the continuous-laser approach in this work is better suited to magnetic features which exhibit higher magnetic noise power densities, allowing higher laser powers and higher subsequent photoluminescence (PL) intensities. This would suggest that the NV system in this implementation is actually progressively less sensitive when used to image weaker (oscillating) magnetic features. There doesn't appear to be any analysis of what sensitivity is required here, and whether this technique is likely to be superior to a more traditional pulsed-laser relaxometry measurement technique.

As mentioned by the Reviewer, the laser power needs to be properly adjusted to optimize the imaging contrast and the NV system is less sensitive when used to image weak magnetic noise. We believe that pulsed-laser relaxometry faces similar limitations. In a single- τ imaging procedure, this is the delay time τ between laser pulse excitations which needs to be properly adjusted. For the detection of a weak magnetic noise, this delay has to be long, typically on the order of T_1 . As a consequence, the duty cycle of optical excitation needs to be reduced, resulting in a decreased PL signal by a factor T_L/T_1 , where T_L is the duration of the laser pulses. To optimize spin readout contrast, the laser pulse duration is commonly set to $T_L=300$ ns with a power close to the saturation power (P_{sat}) of the optical transition. Considering $T_1=100$ μs , the PL signal is thus decreased by two orders of magnitude compared to continuous laser illumination. Using our measurement procedure, the cw laser power has to be decreased to $0.01 \cdot P_{\text{sat}}$ in order to be sensitive to weak magnetic noise (see Fig. 1) leading to similar PL signals. As a result, we believe that the two approaches are likely to offer similar performances. This discussion has been added in section II of the supplementary information.

An in-depth analysis of the magnetic noise sensitivity would require a precise study of (i) the PL quenching effect with a calibrated magnetic noise source and (ii) its evolution with the optical pumping power. This will likely require to introduce more sophisticated models of the NV defect's photodynamics than the one introduced in our work. This is undoubtedly an interesting research direction for future works but we believe that it is out of the scope of the present paper, whose main result is the first demonstration that non-collinear spin textures can be imaged through magnetic noise sensing. Although

not quantitative, this all-optical imaging mode enables to localize such spin textures with high spatial resolution in compensated magnetic materials such as antiferromagnets, which produce weak static magnetic fields. As such, it is a complementary imaging mode to the one relying on PL quenching induced by large “off-axis” magnetic fields [Tetienne *et al*, New J. Phys. **14**, 103033 (2012)], which can be used for studying the physics of spin textures in ferromagnetic systems [*e.g.* Gross *et al*, Phys. Rev. Mater. **2**, 024406 (2018)].

2. The use of PL intensity as a proxy for T_1 is only suitable for NV systems where there is no significant extrinsic T_1 modification. For near-surface NV centres this is known to be problematic, as charge-trapping/transfer and energy resonance effects (eg. FRET) can also modulate the NV PL signature when an NV is brought into close proximity with a sample surface. Similarly, the reflectivity of the sample can also cause significant PL modulation, if the actual laser intensity at the NV is affected. I suggest that this be recognised in the paper, even if the authors believe it to be a mitigated issue in the NV-tip implantation under review.

We agree with the Reviewer. The PL signal of the NV defect can indeed be modified in close proximity with a sample surface. As suggested by the Reviewer we have clearly indicated these effects in the new version of the manuscript, with additional references to the papers by Tisler *et al*. [Nano Lett. **13**, 3152 (2013)] and by Bluvstein *et al*. [Phys. Rev. Lett. **122**, 076101 (2019)]. In our experiments, the NV defect is scanned at a fixed distance over a magnetic material with a uniform surface, *i.e.*, without any detectable topographic features in the AFM scans. As a result, we do not expect local modifications of the PL intensity induced by FRET, charge state conversion effects, or variations of the sample reflectivity during a scan. Furthermore, we believe that our control magnetometry experiments clearly shows that the observed PL quenching lines are located at the position of domain walls in the SAF sample (Fig. 2d), while T_1 measurements confirm the magnetic noise origin of the PL quenching.

3. While the NV-tip system under review is commercially available, it is not clear what is known about the NV-tip distance. More information (at least what is known) should be put in the manuscript, regarding NV orientation, depth from tip and tip radius. This is needed to better understand what is meant by the ‘flying distance’ of the NV spin sensor (page 5).

We thank the Reviewer for their insightful remark. These important details were indeed missing in our manuscript. The distance d_{NV} between the scanning NV sensor and the sample surface was independently calibrated following the procedure detailed in Hingant *et al* [Phys. Rev. Appl. **4**, 014003 (2015)], leading to $d_{NV}=79\pm 5$ nm. The NV defect orientation was measured by recording the ESR frequency as a function of the amplitude and orientation of a calibrated magnetic field. We obtained spherical angles ($\theta = 58 \pm 1^\circ$, $\phi = 103 \pm 1^\circ$) in the laboratory frame of reference. Finally, the diameter of the scanning diamond tip is around 200 nm in order to act as an efficient waveguide for the PL emission of the NV defect.

All of this additional information has been added to the revised SI document (section I.B).

4. In Figure 3 (f-g) a series of domain wall images are recorded using different laser powers. Here it is shown that the normalised PL contrast is increased for lower laser powers. However it is not obvious from these images or otherwise whether the actual signal to noise ratio is improved here. With a lower contrast comes a lower PL intensity (therefore signal intensity). Is there a better way to plot this data, as a proper signal/noise (SNR) analysis? The use of ‘normalised PL’ in most plots appears to obfuscate this issue. Should not there also be more analysis of SNR in this context throughout the manuscript?

For all the PL images shown in the manuscript, the signal is normalised by taking a reference value of the PL averaged in a uniform bright area of the scan. The observed PL quenching contrast could be modified by a background PL signal emanating either from the tip or the sample. Such effects cannot explain the observed reduction of the quenching contrast when the optical power increases [Fig. 3(f-h)]. Indeed, a saturation curve of the PL signal recorded with the NV defect placed in proximity of the SAF sample shows that the signal-to-background ratio is improved when the optical power increases. This saturation curve has been added in the supplementary information (section I.B).

In these experiments, photon shot noise is the main source of noise in the measurement of the PL signal. The signal to noise (SNR) ratio then scales as $(R\Delta t)^{1/2}$, where R is the PL emission rate and Δt the acquisition time per pixel. For a fixed acquisition time, the SNR indeed decreases with the optical illumination power. For all the experiments shown in the manuscript, the

PL fluctuations induced by photon shot noise remain however smaller than the observed PL quenching contrast. Following the suggestion of the Reviewer, the values of R and Δt have been added in the caption of Figure 3. We didn't find a better way to plot the results, that would clearly illustrate the reduction of PL quenching contrast with the optical power.

5. In figure 4 the origin of domain wall noise spectra is shown, including the fact that the domain walls exhibit a much lower spread of magnetic noise intensities – towards DC. This suggests to me that an AC technique, such as a decoherence (T_2) imaging mode may give similar if not superior data for these systems. Is this a universal characteristic of these SAF systems and do the authors still believe that relaxometry gives the best imaging modality?

This idea is very interesting and could maybe be applied to our SAF system but we did not try it. However, the T_2 imaging mode might not be as effective as expected because in this case the NV center is sensitive to low frequency modes lying in the kHz-MHz range [Tetienne *et al*, Nano Lett. **16**, 326 (2016)]. Such modes will indeed exist in the domain walls, in contrast to the domains for which the gap is too large. However, NV centers are also mainly sensitive to the modes with wave vectors such that $k = 1/d_{NV}$. For frequencies in the range probed by T_2 experiments, the wave vectors of the spin waves might actually be too small to be probed with scanning decoherence microscopy.

6. One of the apparent limitations of the NV-tip magnetometry in the context of SAF systems is the common need for magnetic field biasing. In this manuscript a zero-bias relaxometry measurement is performed, I assume primarily to avoid perturbing the sample magnetization. However while this is a feature of the measurement, it would also appear to be a significant limitation. Meaning that only ~3GHz magnetic signals can be detected. As this manuscript is intended to pave the way to future such measurements, this limitation should be recognised and discussed. Is it perhaps useful to suggest the development of alternative control of the NV spin transition frequency, such as controlled-strain NV-tip systems?

We did not apply magnetic field biasing to the SAF samples because this functionality is not available yet in our room-temperature magnetometry setup, in which we can only apply few mT. Nevertheless, there is no issue with applying moderate magnetic fields of several tens of mT to the SAF samples since the RKKY coupling is on the order of 100 mT (see Legrand *et al*, Nat. Mater. **19**, 34 (2020), Supplementary Note 1). More generally, antiferromagnetic spin textures are robust against external magnetic fields. Consequently, it should be possible in future to use the NV sensor as a noise spectrometer by applying a bias magnetic field to antiferromagnetic materials.

7. My reading of this manuscript is that there is a strong need for imaging of net-zero magnetization features. However there is no demonstration of such features here, nor a description of the likely sensitivity required to see such features, or what kinds of SAF devices would exhibit such features?

The spin textures which we observe in the SAF samples are indeed not completely compensated because of the small variations of their profiles in the top and the bottom layer, as indicated in the supplementary information. However, the net magnetic moment here is nevertheless very small. As a result, the imaging with MFM of these antiferromagnetic domain walls, antiferromagnetic spirals and antiferromagnetic skyrmions, is very challenging and requires to work in vacuum with optimized tips. In contrast, our relaxometry-based imaging technique is easy to implement and the only parameter requiring tuning is the laser power. In our work, we have chosen to study an antiferromagnetic system featuring tiny uncompensated magnetic moments - the synthetic antiferromagnet – in order to perform standard magnetometry measurements (static fields) as control experiments. This is a very important point for proving that PL quenching is indeed localized at domain walls (see Fig. 2). We note that the existence of gapless spin wave modes inside domain walls is not specific to SAF systems but very general both in ferromagnets [Garcia-Sanchez *et al*, Phys. Rev. Lett. **114**, 247206 (2015)] and antiferromagnets [Flebus *et al*, Phys. Rev. B **98**, 180409 (2018)].

Reviewer #2 (Remarks to the Author):

In this paper, Finco et. al. present single spin relaxometry as a tool to image non-collinear antiferromagnetic textures. Magnetic noise emanating from non-collinear antiferromagnetic textures is detected as a decrease in photoluminescence signal under continuous laser exposure. The efficiency of this method is highlighted by imaging antiferromagnetic textures like domain walls, spin spirals, and antiferromagnetic skyrmions as realized in synthetic antiferromagnets.

The experimental observations presented here corroborates relaxometry-based imaging as a proof-of-concept following a recent theoretical proposal [Physical Review B 98, 180409 (2018)]. The paper will certainly be of interest to researchers working in this field as it extends the use of relaxometry-based imaging to antiferromagnetic materials which are promising for next-generation spintronics.

We thank the Reviewer for their positive appreciation of our work and provide answers to their remarks below.

In regard to the results presented, I believe a few aspects need to be addressed:

*) For the case of domain walls, when the NV is above the uniform domain at x_d in Fig. 4a, the authors suggest that the Gilbert damping value of 0.1 broadens the magnon spectral density in frequency and thus couples to the NV. First, I do not see any justification/measurement that calibrates the value to alpha to be 0.1. Second, does two-magnon noise play a role in this case - when the NV ESR falls within the magnon spectrum gap as shown in Fig. 4b?

Concerning the first point: Sputtered ultrathin Co films on Pt are known to possess relatively large values of the Gilbert damping constant, i.e., $\alpha \geq 0.1$. We have performed Brillouin light scattering spectroscopy measurements of single Co layers in Pt (8 nm)/Co (1.5 nm)/Ru (4 nm) sandwiches in which spectral linewidths give an estimate of the Gilbert damping. These films were deposited under the same conditions as the synthetic antiferromagnets studied in the manuscript. An example is presented in Figure R1 below, where BLS spectra were investigated as a function of an applied in-plane magnetic field H_x . Above saturation in the film plane, the uniform resonance frequency is given by the Kittel relation

$$\omega = \sqrt{\omega_0(\omega_0 + \omega_M)}$$

where $\omega_0 = \gamma_0(H_x - H_K)$ represents the frequency associated with the difference between the in-plane applied field and the effective perpendicular anisotropy field, H_K , and $\omega_M = \gamma_0 M_s$, where M_s is the saturation magnetization. A fit of the experimental data to this equation is shown in Fig. R1(a). Assuming Gilbert damping, the full width at half maximum is given by

$$FWHM = 2\alpha\omega_0 \frac{\partial\omega}{\partial\omega_0} = \alpha \frac{\omega_0(2\omega_0 + \omega_M)}{\sqrt{\omega_0(\omega_0 + \omega_M)}}$$

From the spectral line in Fig. R1(b,c), we have $\omega \approx 11$ GHz and $FWHM \approx 2$ GHz. By assuming $M_s = 1.4$ MA/m, we obtain an estimate of $\alpha \approx 0.17$, which is in line with our assumption. This is also consistent with previous estimates based on time-resolved MOKE experiments on the transient magnetization dynamics in Pt/Co/AlOx where values of $\alpha = 0.11$ to 0.28 were found [A. J. Schellekens et al, Appl. Phys. Lett. **102**, 082405 (2013)].

Fig. R1. (a) Dependence of resonance frequency with in-plane applied field in a Pt/Co (1.5 nm)/Ru film, measured with BLS. (b) Example spectrum at high fields, illustrating the Stokes and anti-Stokes peaks. (c) Zoom on the anti-Stokes peak in (b), where the central frequency can be seen to be ~ 11 GHz with a full-width-half-maximum of 2 GHz.

Concerning the second point: Two-magnon scattering refers to a wave vector conversion processes by which, for example, the uniform mode ($k = 0$) mode can scatter into a finite wave vector mode ($k \neq 0$) *at the same frequency*. This process can take place in thin magnetic films in which translational invariance in the film plane is broken, e.g., by film roughness or defects. While it plays a role in magnetic relaxation (or decoherence at least), two-magnon scattering is not a process that can be described by Gilbert damping and therefore is not accounted for explicitly by the simulations discussed in Fig. 4. Nevertheless, because two magnon scattering describes conversion between modes of the same frequency, it cannot lead to an additional noise contribution to the NV spin relaxation if a mode at the ESR frequency does not exist in the first place. In other words, no scattering processes can result in a mode at the ESR frequency if it remains in the gap as shown in Fig. 4b.

**) For the case of imaging antiferromagnetic skyrmions, the authors suggest that the reason that NV senses the skyrmion (inspite of the fact that its eigenspectrum does not have a mode at 2.87 GHz) by means of scattered magnons against the skyrmion. The supplementary material argues this by evaluating the spinwave dispersion of the synthetic antiferromagnet in the uniform state which does have modes at the NV ESR frequency. The conclusion based on this argument is not very convincing.*

Fig. R2 shows the simulated excitation spectra of an isolated skyrmion in the biased synthetic antiferromagnetic structure, using the same parameters as those described in Section III of the Supplementary Information. The spectra were computed from the transient response of the skyrmion to a pulsed magnetic field along the z axis, perpendicular to the film plane. In the top panel, we present the case where a smaller damping constant of $\alpha = 0.01$ was used, which is helpful to resolve the modes. We observe that the mode frequencies lie in the range below the ESR frequency, with the main breathing mode frequency occurring at around 1 GHz. Broadening of the spectral line can be seen for the more realistic case of $\alpha = 0.1$, where the breathing mode peak is still visible while the other modes are broadened. In this case, however, broadening only trickles into the vicinity of the ESR frequency.

Fig. R2. Power spectral density (PSD) of skyrmion oscillations in a synthetic antiferromagnet with a bias layer for two different values of the Gilbert damping constant. The PSD is computed from the transient response of the m_z component of the total magnetization as result of a 2-mT magnetic field pulse applied along the z direction. f_{NV} indicates the ESR frequency.

Based on these results, it is unlikely that skyrmion modes contribute directly to the noise spectrum at the ESR frequency. It is therefore more like that the skyrmion acts as a scattering centre for thermally-excited spin wave modes. Our simulations bear this out, since we can observe an increase in the noise intensity around the vicinity of the skyrmion core, as shown in Fig. S5(d) and S5(f). We therefore believe that our interpretation is a plausible one.

The results of this paper are indeed novel and will surely be of interest to the researchers working in this field. However, with proper clarifications provided for the points mentioned above, I do think that this manuscript satisfies the criterion for publication in Nature Communications.

We thank the Reviewer for their insightful remarks. We believe we have addressed the issues raised in full.

Reviewer #3 (Remarks to the Author):

Finco et al. use optical measurements of the spin-relaxation rate of a scanning NV center in a diamond tip to image domain boundaries in antiferromagnetic structures with nanoscale resolution. Due to a small or absent net magnetization, antiferromagnets are notoriously challenging to image via magnetometry. Finco et al. show that antiferromagnetic domain boundaries in fact generate strong local magnetic noise that can be detected via the NV relaxation rate. They characterize this rate by its impact on the NV photoluminescence, thus avoiding the necessity of using microwaves. This makes it a straightforward method that starts to approach the ease-of-use of the magnetic force microscope (MFM) while going beyond MFM in its ability to detect magnetic fields that fluctuate rapidly (on the GHz scale). They image a variety of spin structures and provide a theoretical explanation for the microscopic origin of the detected magnetic signals. This work pushes the state-of-the-art in NV magnetometry and provides new insight into magnetism. Because it highlights the potential of using NV-magnetometry for general materials characterization, it should be of broad interest in the condensed-matter physics and materials science communities. The data are well presented and the conclusions are supported by the data and their analysis. It should be accepted in Nature communications.

We thank the reviewer for his very positive report and address his comments below.

I have some minor comments:

The authors could elaborate on the challenges for extracting T_1 quantitatively from photoluminescence maps. In particular since the technique relies on striking a rather precise balance between illumination power (a control parameter) and relaxation time T_1 (not a control parameter). This may limit the material systems to which this technique can be applied and extend the time required for calibration measurements. Can the authors comment on the shortest T_1 they can determine via this technique?

The scanning relaxometry mode which we propose here is *not* a quantitative measurement mode. We cannot extract quantitatively T_1 from our PL maps. In this aspect, the relaxometry mode is similar to the PL quenching mode which is used to localize magnetic textures producing large stray fields in ferromagnets. The strength of this imaging mode is that it provides a fast and simple way to localize the non-collinear textures in the sample since they are sources of magnetic noise. However, once their position is known, quantitative local measurements of T_1 can be achieved.

The balance between T_1 and the laser power is actually not so difficult to obtain, as illustrated by our experiments on three different samples. The measured T_1 values in the bright (dark) areas are respectively 120 μs (22 μs), 920 ns (400 ns) and 9 μs (870 ns) on the domain-wall sample, the spiral sample and the skyrmion sample, covering already a wide range of 3 orders of magnitude (Fig 3 and SI Figs. S9, S10).

Can the authors comments on the limit in optical power that can be applied? (at some point a high optical excitation power should cause the NV to reside most of the time in the electronic excited state)

Only very short values of T_1 can be probed through the PL variation if the applied optical power is large. In the two-level model described in Fig. 1 and in SI (section II) the value of T_1 where the PL drop occurs corresponds to the inverse of the polarization rate. At the saturation power, it becomes on the order of the lifetime of the metastable state. In addition, if we increase too much the laser power, we could also have issues with charge conversion of the NV state.

It is unclear how the data (in e.g. Fig. 2b) are normalized. Please explain. I also wonder if the normalization procedure is prone to artefact: In particular, it seems to me that a spatial change in background (non-NV) photoluminescence would also give a change in contrast. Can the authors comment on this? And why do the authors plot the ratio and not the difference between the two PL rates?

For all the PL images shown in the manuscript, the signal is normalised by taking a reference value of the PL averaged in a uniform bright area of the scan. We plot the ratio of the PL rate with the reference value in order to be able to compare easily by eye the contrast obtained in the PL images. A change in PL background would indeed change the contrast but the surface of our sample is very uniform and the background PL is small, as illustrated by the saturation curve which we added in the SI, Fig. S2.

Since the authors chose a material system in which the stray fields of boundaries are large enough to be imaged by static magnetic field imaging, it is not immediately obvious that PL-based relaxometry can access information that traditional vector magnetometry through ODMR cannot. Can the authors be explicit about what new information can be extracted?

When it is possible to perform the usual quantitative magnetometry measurements on the sample, the relaxometry mode provides indeed less information than a stray field map. However, the acquisition time in the relaxometry mode is about 5 times faster. In addition, when T_1 is very short, it becomes challenging to measure good enough ODMR spectra to actually perform the vector magnetometry experiment. This is for instance the case for the spiral sample, on which we were not able to record a stray field map, whereas the relaxometry mode allows us to directly image the spin spiral configuration. In a more general context, the relaxometry-based imaging mode could be particularly useful for studying magnetic order in compensated systems, where stray fields are vanishingly small but thermal spin waves are amply present under ambient conditions. Ultimately for a proof-of-concept study such as ours, it was imperative to be able to validate the relaxometry measurements with stray field measurements.

Can the authors comments on the presence of domain structures in Figure 3h? According to the presented theory in Figure 1, shouldn't there be no noticeable difference in PL between $T_1=22 \mu\text{s}$ and $T_1=860 \mu\text{s}$?

There is indeed a faint domain wall pattern in Fig 3h, but the contrast here is reduced to 1 % in average (to compare with the 7 % contrast at low power in Fig 3f, see details in SI, section IV.B). Looking at the curve in Fig. 1, one has to consider the difference in PL between $T_1 = 22 \mu\text{s}$ on the domain wall and $T_1 = 120 \mu\text{s}$ on the domain for high laser power. The curve indicates that we should not expect a noticeable contrast, which is the case.

A different order of the labels in Figure 4 would be clearer

The figure has been rearranged to improve this.

Reviewers' Comments:

Reviewer #1:

Remarks to the Author:

The authors have responded well to all reviewers' questions and criticisms. This has resulted in an improved manuscript (and SI) and it is certainly of high quality and suitable for publication. The article demonstrates application of an existing measurement system (NV-diamond pillars) to a difficult magnetic imaging problem. This will be of significant interest to the journal's readership, and the manuscript (with SI) is well presented and now comprehensive in detailing what was shown and achieved. The weak aspects of this manuscript remain, in that it doesn't demonstrate imaging of any new material features, it is not quantitative and its utility may be limited by sample specifics. For these reasons I believe the manuscript's suitability for Nat. Comms. is still borderline. But on balance I do recommend it for publication.

Reviewer #2:

Remarks to the Author:

The authors have addressed all my concerns and questions. With that, I would recommend the manuscript for publication in Nature Communications.

Reviewer #3:

Remarks to the Author:

The authors have addressed my requests. I recommend publication